# Serum $\beta_2$-microglobulin levels in Coronavirus disease 2019 (Covid-19): Another prognosticator of disease severity?

Walter Conca[1,2,3,4☯‡]*, Mayyadah Alabdely[1‡], Faisal Albaiz[1‡], Michael Warren Foster[2‡], Maha Alamri[1], Morad Alkaff[5], Futwan Al-Mohanna[3,4], Nicolaas Nagelkerke[6☯¤‡], Reem Saad Almaghrabi[1]

1 Department of Medicine, King Faisal Specialist Hospital & Research Centre, Riyadh, Saudi Arabia,
2 Department of Executive Health Medicine, King Faisal Specialist Hospital & Research Centre, Riyadh, Saudi Arabia, 3 Department of Cell Biology, King Faisal Specialist Hospital & Research Centre, Riyadh, Saudi Arabia, 4 College of Medicine, Alfaisal University, Riyadh, Saudi Arabia, 5 Department of Pathology and Laboratory Medicine, King Faisal Specialist Hospital & Research Centre, Riyadh, Saudi Arabia,
6 Department of Community Medicine, United Arab University, Al Ain, United Arab Emirates

☯ These authors contributed equally to this work.
¤ Current address: Goetzens, Austria (retired)
‡ MA, FA and MWF also contributed equally to this work. WC and NN are joint senior authors on this work.
* wconca@kfshrc.edu.sa, concawalter6@gmail.com

**Data Availability Statement:** All relevant data are within the manuscript and its Supporting information files.

## Abstract

$\beta_2$-microglobulin ($\beta_2$-m), a 11.8 kDa protein, pairs non-covalently with the α3 domain of the major histocompatibility class (MHC) I α-chain and is essential for the conformation of the MHC class I protein complex. Shed $\beta_2$-m is measurable in circulation, and various disorders are accompanied by increases in $\beta_2$-m levels, including several viral infections. Therefore, we explored whether $\beta_2$-m levels could also be elevated in Coronavirus disease 2019 (Covid-19) and whether they predict disease severity. Serum $\beta_2$-m levels were measured in a cohort of 34 patients infected with SARS-CoV-2 on admission to a tertiary care hospital in Riyadh, Saudi Arabia, as well as in an approximately age-sex matched group of 34 uninfected controls. Mean $\beta_2$-m level was 3.25±1.68 mg/l (reference range 0.8–2.2 mg/l) in patients (mean age 48.2±21.6) and 1.98±0.61 mg/l in controls (mean age 48.2±21.6). 17 patients (mean age 36.9± 18.0) with mean $\beta_2$-m levels of 2.27±0.64 mg/l had mild disease by WHO severity categorization, 12 patients (mean age 53.3±18.1) with mean $\beta_2$-m levels of 3.57±1.39 mg/l had moderate disease, and five patients (of whom 2 died; mean age 74.4 ±13.8) with mean $\beta_2$-m levels of 5.85±1.85 mg/l had severe disease ($P < = 0.001$, by ANOVA test for linear trend). In multivariate ordinal regression $\beta_2$-m levels were the only significant predictor of disease severity. Our findings suggest that higher $\beta_2$-m levels could be an early indicator of severity of disease and predict outcome of Covid-19. As the main limitations of the study are a single-center study, sample size and ethnicity, these results need confirmation in larger cohorts outside the Arabian Peninsula in order to delineate the value of $\beta_2$-m measurements. The role of $\beta_2$-m in the etiology and pathogenesis of severe Covid-19 remains to be elucidated.

**Funding:** The author(s) received no specific funding for this work.

**Competing interests:** The authors have declared that no competing interests exist.

**Abbreviations:** AIDS, aquired immunodeficiency syndrome; ANOVA, analysis of variance; ARDS, acute respiratory distress syndrome; CBC, complete blood count; CMV, cytomegalovirus; COPD, chronic obstructive pulmonary disease; Covid-19, Coronavirus disease 2019; CRP, C-reactive protein; EBV, Epstein-Barr virus; eGFR, estimated glomerular filtration rate; f, female; FGF, fibroblast growth factor; G-CSF, granulocyte colony stimulating factor; GM-CSF, granulocyte monocyte colony stimulating factor; HIV, human immunodeficiency virus; ICU, intensive care unit; IFN, interferon; IL, interleukin; IL-1Ra, interleukin-1 receptor antagonist; IP-10, interferon γ-induced protein-10; m, male; MCP, monocyte chemoattractant protein-1; MHC, major histocompatibility complex; MIP, macrophage migration inhibitory factor; NK, natural killer; PCR, polymerase chain reaction; PDGF, platelet derived growth factor; SARS-CoV-2, severe acute respiratory syndrome coronavirus 2; SpO2, saturation pulse O2; TNF, tumor necrosis factor; VEGF, vascular endothelial growth factor; WBC, white blood cell; WHO, World Health Organization; $\beta_2$-m, $\beta_2$-microglobulin.

# Introduction

Immune responses to the infection with SARS-CoV-2, the causative pathogen of Covid-19, were first described in China [1–3] and subsequently, in the wake of a global spread, in other countries and ethnicities [4–12]. Covid-19 is preponderantly mild and self-limiting with rapidly developing anti-viral immunity. By contrast, progression to a well characterized cytokine release syndrome resulting in acute lung injury or acute respiratory distress syndrome in critical cases with a high case fatality ratio occurs in many elderly and in the presence of comorbidities, similar to other (seasonal) viral infections [13–15]. Immunosenescence with exhaustion of plasmacytoid dendritic cells, natural killer (NK) cells and cytotoxic CD8+ T cells, which form the frontline of cellular innate and adaptive anti-viral immune defense, would seem a plausible explanation of the inadequacy of the aging immune system to clear this virus efficiently [16–19]. In severe or critical disease, a markedly impaired interferon (IFN) type I response was associated with a persistent blood viral load and florid inflammation, which in some patients was also related to the presence of autoantibodies against IFN type I [20, 21]. Host-pathogen interactions, distinct immunotypes and immune signatures are determinants of pathogenesis and correlate with outcomes of Covid-19 [11, 12, 22]. A fulminant cytokine release syndrome or storm, including common protagonists of inflammation such as IL-1β, IL-1Ra, IL-2, IL-6, IL-7, IL-8, IL-9, IL-10, basic FGF, G-CSF, GM-CSF, IFN-γ, IP-10, MCP-1, MIP-1α, MIP-1β, PDGF, VEGF and TNF-α, is made responsible for life-threatening respiratory failure and multi-organ dysfunction or failure [1]. With regard to outcome, the principal, known predictors of mortality are advanced age, comorbidities such as diabetes, hypertension, cardiac disease, chronic lung disease, chronic kidney disease, cerebrovascular disease, dementia, mental disorders, immunosuppression, obesity and cancer, and laboratory parameters, *i.e.* CRP, LDH, cardiac troponin I, ferritin, D-dimers, and raised levels of IP-10, IL-10, IL-1Ra, IL-6, as well as lymphopenia [23–29]. In combination with the clinical presentation and radiographic findings, some of these variables have been used to develop various prognostic models for risk stratification of patients admitted to hospital with Covid-19, but a universally accepted and applicable scoring system has as yet not been established [30].

With the intention to explore whether other plausible markers could be clinically useful for assessing and predicting the response of the immune system to SARS-CoV-2, we determined the blood levels of $\beta_2$-microglobulin ($\beta_2$-m) in patients with Covid-19 who were admitted to a tertiary care hospital in Riyadh, Saudi Arabia. For comparison, $\beta_2$-m levels were also measured in an approximately age-sex matched healthy control group.

The hypothesis that $\beta_2$-m measurements could be relevant was suggested by previous findings of abnormal $\beta_2$-m levels in a variety of viral infections, including those caused by EBV, CMV and influenza virus [31]. $\beta_2$-m concentrations were highest in patients with CMV disease (6.5±2.0 mg/l), infectious mononucleosis (4.8±1.7 mg/l) and influenza A (4.2±1.9 mg/l). In HIV infection, a serum $\beta_2$-m level of >3 mg/l predicted the development of AIDS within a period of 36 months [32]. Intriguingly, IL-1Ra, a biomarker of disease severity in Covid-19 [27], correlated positively with serum $\beta_2$-m levels and negatively with circulating CD4+ T lymphocyte counts in patients during different stages of AIDS, suggesting a shared regulatory pathway among these two disparate viral infections [33]. The pathophysiologic mechanisms which are widely accepted as the cause of elevated blood levels of $\beta_2$-m under these circumstances are an accelerated rate of shedding or dissociation of $\beta_2$-m from the MHC class I α-chain at the cell surface of immune and non-immune cells, which relates to the cardinal function of the MHC class I system of presenting viral antigenic peptides to cytotoxic CD8+ T cells. The ensuing death of antigen presenting cells releases cellular

components collectively designated as "damage-associated molecular patterns" (DAMPs) that evoke an innate and adaptive immune response which may harm the host if regulatory mechanisms fail [34–37]. After engagement of the T-cell receptor with the trimeric MHC class I complex, the α-chain dissociates from $\beta_2$-m, then is internalized and degraded, while $\beta_2$-m is released from the cell surface [38, 39]. Apart from this mechanism of disintegration, there is evidence that up-regulation of $\beta_2$-m synthesis also occurs at the transcriptional level in response to stimulation by a variety of cytokines, such as IFN type I, TNF-α and IL-1$\beta$ [40–42].

## Methods

### Setting

We conducted an observational study at the King Faisal Specialist Hospital & Research Centre (KFSHRC), a large (~1.000.000 outpatient visits/year, 1.600 beds, ~1.000 doctors and ~13.500 employees), non-profit, tertiary referral hospital, located in the city center of Riyadh, Saudi Arabia, where patients are referred from other hospitals from across Saudi Arabia and adjacent regions. Participants were included consecutively in the period from 14 March to 8 April 2020. All patients were Saudi nationals except two European expatriates. The Saudi nationals were self-referrals, as these participants were long-term patients and therefore have direct access to health care in this hospital. The two European patients were admitted after special arrangements. The diagnosis of Covid-19 was suspected clinically and confirmed through the detection of SARS-Cov-2 in a nasopharyngeal sample with specific PCR (RealStar®SARS-CoV-2 RT-PCR Kit RUO altona-diagnostics, Germany), which was performed in the Section of Medical Microbiology of the Department of Pathology and Laboratory Medicine at KFSHRC. Upon testing positive, the patients were admitted and isolated in negative pressure rooms. Severity of disease, *i.e.* mild, moderate, severe or critical, was determined on admission and modified according to the clinical course using the WHO categories of Covid-19 disease severity [43]. These are defined as follows: *mild* disease, no evidence of viral pneumonia or hypoxia; *moderate* disease, clinical signs of pneumonia (fever, cough, dyspnea, fast breathing), but not severe pneumonia, including SpO2 ≥ 90% on room air, cautioning that a SpO2 of >90–94% on room air is abnormal in a patient with normal lung and can be an early sign of severe disease; *severe* disease, clinical signs of pneumonia (fever, cough, dyspnea, fast breathing) plus one of the following: respiratory rate >30 breaths/min, severe respiratory distress or SpO2 <90% on room air. Chest imaging (radiograph, CT scan, ultrasound) may assist in diagnosis and identify or exclude pulmonary complications in moderate and severe disease. *Critical* disease, not distinguished from severe disease in our analyses (because of small numbers), is defined by onset (within 1 week of a known clinical insult, *i.e.* pneumonia, or new or worsening respiratory symptoms), chest imaging (radiograph, CT scan or ultrasound showing bilateral opacities, not fully explained by volume overload, lobar or lung collapse, or nodules), origin of pulmonary infiltrates (respiratory failure not fully explained by cardiac failure or fluid overload by objective assessment, *e.g.* echocardiography) and oxygenation impairment (mild, moderate or severe acute respiratory distress syndrome). One patient with Covid-19 had $\beta_2$-m levels between 19.85 and 40.35 mg/l, but was on intermittent hemodialysis and was therefore excluded from our analyses. For comparison, we also determined $\beta_2$-m levels and other parameters in a group of approximately age-sex matched controls without evidence of any infection who were recruited from hospital personnel, trainees or patients. The study was approved by the Hospital's ethics committee, the Research Advisory Council (RAC No: 2201052) and written informed consent was obtained from all subjects.

## Data sources

Clinical, laboratory and radiographic data were handled *via* the electronic medical records system (PowerChart; Cerner, USA), and included participants' demographic details, vital signs, admission notes with chief complaint(s), history of present illness, previous diagnoses, medications, laboratory tests, radiography, and progress notes. These data were extracted from the electronic medical records, deposited and further processed using REDCap [44]. Within this project, each participant patient was assigned a unique research-specific ID number that was password-protected and accessible to one of the investigators (R.S.A). Data were exported from REDCap into a Microsoft Excel spreadsheet which is included as (S1 File. DataSetCovid_Saudia.Excel).

## Variables assessed

For each patient with Covid-19, we obtained and recorded electronically the following data at the time of admission to the hospital: age, sex, nationality, vital signs including SpO2, presenting symptom(s), comorbidities, medications, laboratory tests and chest X-ray. Laboratory investigations included complete blood count (CBC), absolute counts of CD3$^+$, CD3$^+$CD4$^+$ and CD3$^+$CD8$^+$ T cells, CD19$^+$ B cells, CD56$^+$CD16$^+$ NK cells, levels of $\beta_2$-m, ferritin, D-dimer, CRP, estimated glomerular filtration rate (eGFR, CKD-EPI equation) [45]. In the control group, after exclusion of an infectious disease, the same hematologic and biochemical parameters were measured. All variables are provided in the (S1 File. DataSetCovid_Saudia. Excel).

## Quantification of $\beta_2$-m, ferritin, D-dimer and CRP levels

The tests for $\beta_2$-m, ferritin, D-dimer and CRP were performed within two days of admission in the Medical Laboratory of the Department of Pathology and Laboratory Medicine at KFSHRC. Serum $\beta_2$-m levels were quantified using an immunoturbidometric assay with a latex-bound rabbit polyclonal anti-$\beta_2$-m antibody on Roche/Hitachi cobas® c system (TINA-QUANT® $\beta_2$-microglobulin). The measuring linear range of this particular assay is 0.2–8.0 mg/l, and the reference range is 0.8–2.2 mg/l. Serum levels of ferritin were determined by electrochemiluminescence immunoassay (Elecsys Ferritin®) using streptavidin-coated microparticles, biotinylated mouse monoclonal anti-ferritin antibody and ruthenium-complex-labeled mouse monoclonal anti-ferritin antibody on the Roche/Hitachi cobas® e 801 immunoassay analyzer (measuring range: 0.50–2000 µg/l; reference range for men: 30–400 µg/l; for women: 13–150 µg/l). D-dimer levels were determined in plasma with an immunoturbidometric assay (STA®-Liatest® D-DI PLUS) using latex microparticles coated with two different mouse monoclonal anti-D-dimer antibodies (normal level < 0.5 µg/ml FEU) and analyzed on the STA-R® Max2 instrument. CRP levels were measured in serum using an immunoturbidometric assay with latex particles coated with mouse monoclonal anti-CRP antibody (CRPHS®) on the Roche/Hitachi cobas® c system (measuring range: 0.15–20.0 mg/l). For this high-sensitivity CRP assay, levels >10 mg/l indicate systemic inflammation.

## Phenotyping of circulating lymphocytes

Flow cytometry for lymphocyte subsets was performed on heparinized whole blood using BD Multitest™ 6-color TBNK reagent and a six-color direct immunofluorescence assay with BD Trucount™ tubes on BD FACSCanto™ II flow cytometer instrument (Becton Dickinson Biosciences, San Jose, CA, USA) using standard quality control and instrument settings [46]. At least 5,000 lymphocytes were acquired and analysis was performed using BD FACSDiva™ software

version 10.0 (BD Biosciences). All results were expressed as absolute counts and percentages of mature T, B, and NK lymphocyte populations as well as CD4$^+$ and CD8$^+$ T-cell subset ratios in peripheral blood.

## Statistical analysis

Tabulations, analysis of variance (ANOVA) including linear trend tests, ordinal logistic regression (proportional odds model, stepwise manually, backward selection, $p$-value based, $p$\_out = 0.05), and Pearson's correlation coefficients were used. A significance level (2-tailed) of 0.05 was used throughout. As the distribution of CRP levels was highly skewed, its values were 10log transformed. Analyses were carried out with SPSS v.22 (IBM SPSS Statistics for Windows, Version 22.0. Armonk, NY: IBM Corp). The raw data used for statistical analysis are included in the (S1 File. DataSetCovid_Saudia.Excel).

## Results

### Patients' characteristics

Demographic characteristics, main comorbidities, clinical manifestations, medications and vital signs on presentation are shown in Table 1. Individual data of each patient are presented separately in a (S1 Table). 34 consecutive participants (mean age 48.2 ± 21.6; 12 m, 22 f) presented to the emergency department with one or more of the following chief complaints (in descending order of frequency): fever, dry cough, sore throat, rhinorrhea, fatigue, headache, diarrhea, anosmia, productive cough, dyspnea, ear pain, ageusia, anorexia, abdominal pain, nausea, emesis, seizure, syncope, myalgia, rash and no symptoms. Fever and dry cough were the principal manifestations in all patients, whereas other symptoms varied among severity groups. One patient who progressed to severe disease was presymptomatic at presentation. Among the patients, comorbidities were as follows: hypertension, diabetes mellitus, dyslipidemia, coronary artery disease, congestive heart failure, atrial fibrillation, chronic obstructive pulmonary disease, cerebrovascular disease, leukemia in remission, colorectal cancer in remission, post-renal transplant, Hodgkin's lymphoma in remission, and hypothyroidism. On admission, 23 participants (68%) were on one or more of the following medications: lisinopril, ramipril, perindopril, losartan, valsartan, amlodipine, diuretics (furosemide/thiazide), metformin, liraglutide, atorvastatin, amiodarone, flecainide, apixaban, rivaroxaban, warfarin, L-thyroxine, fluticasone/salmeterol, montelukast, imatinib, tacrolimus, prednisone, diphenyl-hydantoin, paroxetine, pregabalin, escitalopram.

### Outcome data

Serum $\beta_2$-m levels were raised above reference range in 26 patients (76%) with a mean level of 3.25± 1.68 mg/l. The lowest $\beta_2$-m level of 1.16 mg/l was observed in a 25-year old woman, the highest 8.9 mg/l in a 90-year old man. The clinical impression was that those patients with moderate (12 patients) or severe disease (5 patients) had higher $\beta_2$-m levels. Two patients with severe disease, a 71-year old man and a 90-year old man, did not survive respiratory failure. Their $\beta_2$-m levels on admission of 6.24 mg/l and 8.9 mg/dl, respectively, were the highest observed. In an approximately age-sex matched control group of 34 individuals (mean age 48.2±21.6) without infection, mean $\beta_2$-m level was 1.98±0.61 mg/l. Of interest was the strong correlation (Pearson's r = 0.77, p<0.001) of $\beta_2$-m levels with age in the control group.

Therefore, further statistical analyses (ordinal logistic regression, correlation coefficients) were performed on three groups: group A of 17 patients with mild disease, group B of 12 patients with moderate disease and group C of five patients with severe disease. Patients in

**Table 1. Demographic characteristics, comorbidities, clinical manifestations, medications and vital signs in patients admitted to the hospital with Covid-19.**

| | All Patients (n = 34) | Mild Disease (n = 17) | Moderate Disease (n = 12) | Severe Disease (n = 5) |
|---|---|---|---|---|
| Age (years), mean (±SD) | 48.2 (21.6) | 36.9 (18.0) | 53.3 (18.1) | 74.4 (13.8) |
| Range | 20–90 | 20–79 | 22–78 | 54–90 |
| Age by group, n (%) | | | | |
| <40 | 13 (38.2) | 10 (58.8) | 3 (25) | 0 (0) |
| 40–59 | 9 (26.5) | 4 (23.5) | 4 (33.3) | 1 (20) |
| 60–80 | 10 (29.4) | 3 (17.6) | 5 (41.7) | 2 (40) |
| 80+ | 2 (5.9) | 0 (0) | 0 (0) | 2 (40) |
| Sex, n (%) | | | | |
| Male | 12 (35.3) | 5 (29.4) | 2 (16.7) | 5 (100) |
| Female | 22 (64.7) | 12 (70.6) | 10 (83.3) | 0 (0) |
| Comorbidities, n (%) | | | | |
| Hypertension | 12 (35.3) | 2 (11.8) | 6 (50) | 4 (80) |
| Diabetes | 6 (17.6) | 0 (0) | 3 (25) | 3 (60) |
| Dyslipidemia | 5 (14.7) | 0 (0) | 3 (25) | 2 (40) |
| Coronary Artery Disease | 3 (8.8) | 0 (0) | 0 (0) | 3 (60) |
| Congestive Heart Failure | 3 (8.8) | 0 (0) | 1 (8.3) | 2 (40) |
| Atrial fibrillation | 3 (8.8) | 1 (5.9) | 0 (0) | 2 (40) |
| COPD | 2 (5.9) | 0 (0) | 0 (0) | 2 (40) |
| Cerebrovascular Disease | 1 (2.9) | 0 (0) | 0 (0) | 1 (20) |
| Other | 4 (11.8) | 2 (11.8) | 1 (8.3) | 1 (20) |
| Symptoms at presentation, n (%) | | | | |
| Fever | 23 (67.6) | 10 (58.8) | 10 (83.3) | 3 (60) |
| Dry cough | 23 (67.6) | 9 (52.9) | 10 (83.3) | 4 (80) |
| Sore throat | 14 (41.2) | 9 (52.9) | 5 (41.7) | 0 (0) |
| Rhinorrhea | 12 (35.3) | 8 (47.1) | 3 (25) | 1 (20) |
| Fatigue | 10 (29.4) | 5 (29.4) | 3 (25) | 2 (40) |
| Headache | 8 (23.5) | 3 (17.6) | 4 (33.3) | 1 (20) |
| Diarrhea | 5 (14.7) | 1 (5.9) | 4 (33.3) | 0 (0) |
| Anosmia | 5 (14.7) | 3 (17.6) | 2 (16.7) | 0 (0) |
| Productive cough | 4 (11.8) | 1 (5.9) | 1 (8.3) | 2 (40) |
| Dyspnea | 3 (8.8) | 2 (11.8) | 1 (8.3) | 0 (0) |
| Otalgia | 2 (5.9) | 1 (5.9) | 0 (0) | 1 (20) |
| Ageusia | 2 (5.9) | 1 (5.9) | 1 (8.3) | 0 (0) |
| Anorexia | 2 (5.9) | 1 (5.9) | 1 (8.3) | 0 (0) |
| Nausea | 1 (2.9) | 0 (0) | 1 (8.3) | 0 (0) |
| Vomiting | 1 (2.9) | 0 (0) | 0 (0) | 1 (20) |
| Abdominal pain | 1 (2.9) | 0 (0) | 1 (8.3) | 0 (0) |
| Seizure | 1 (2.9) | 0 (0) | 0 (0) | 1 (20) |
| Syncope | 1 (2.9) | 0 (0) | 0 (0) | 1 (20) |
| Myalgia | 1 (2.9) | 1 (5.9) | 0 (0) | 0 (0) |
| Rash | 1 (2.9) | 0 (0) | 1 (8.3) | 0 (0) |
| Asymptomatic | 1 (2.9) | 0 (0) | 0 (0) | 1 (20) |
| Medications, n (%) | | | | |
| Beta-Blocker | 8 (23.5) | 2 (11.8) | 2 (16.7) | 4 (80) |
| ACEI or ARB | 7 (20.6) | 1 (5.9) | 2 (16.7) | 4 (80) |
| Oral Hypoglycemic | 5 (14.7) | 0 (0) | 3 (25) | 2 (40) |
| Diuretic | 4 (11.8) | 0 (0) | 1 (8.3) | 3 (60) |

*(Continued)*

**Table 1.** (Continued)

| | All Patients (n = 34) | Mild Disease (n = 17) | Moderate Disease (n = 12) | Severe Disease (n = 5) |
|---|---|---|---|---|
| Anticoagulant | 3 (8.8) | 1 (5.9) | 0 (0) | 2 (40) |
| Antidepressant | 3 (8.8) | 2 (11.8) | 1 (8.3) | 0 (0) |
| Immunosuppresant | 2 (5.9) | 0 (0) | 1 (8.3) | 1 (20) |
| Antiepileptic | 1 (2.9) | 0 (0) | 0 (0) | 1 (20) |
| Other | 17 (50) | 6 (35.3) | 8 (66.7) | 3 (60) |
| Vitals at presentation, mean (±SD) | | | | |
| Temperature | 37.1 (0.53) | 37 (0.5) | 37.3 (0.5) | 37.3 (0.8) |
| Heart Rate | 83 (13.7) | 82 (13.3) | 86 (14.7) | 82 (15.1) |
| Systolic Blood Pressure | 124 (15.3) | 119 (16.3) | 130 (12.7) | 126 (13.8) |
| Diastolic Blood Pressure | 76 (10.7) | 75 (10.6) | 81 (11.7) | 70 (1.8) |
| Respiratory Rate | 20 (1.8) | 20 (0.7) | 20 (0.8) | 22 (4.2) |
| O$_2$ saturation | 97% (2.5) | 98% (1.7) | 96% (1.9) | 94% (3.2) |
| BMI | 27.8 (4.3) | 27 (3.5) | 29 (4.7) | 28.2 (6.3) |

Mean age (±SD), sex, comorbidities, symptoms and signs, medications and means of vital signs (±SD) on presentation to the hospital are listed in a cohort of 34 patients diagnosed with Covid-19 (2nd column). The patients were further categorized according to WHO criteria of disease severity as mild (n = 17; 3rd column), moderate (n = 12; 4th column) and severe disease (n = 5; 5th column).

group A had initial mean $\beta_2$-m levels of 2.27 ±0.64 mg/l, group B 3.57 ±1.39 mg/l and group C 5.85 ±1.85 mg/l. Mean age differed among groups, with 36.9± 18.0 in group A, 53.3±18.1 in group B, and 74.4±13.8 in group C. All patients in group B and C had at least one comorbid condition. As in the control group, age and $\beta_2$-m levels were significantly correlated (Pearson's r = 0.71, p<0.001). In addition, $\beta_2$-m levels were significantly correlated with several other risk factors in our patients, notably WBC (r = 0.36), D-dimer (r = 0.54), 10log (CRP) (r = 0.62) and eGFR (r = 0.75). The comparison of all groups, as well as the relationship between $\beta_2$-m levels and age, is graphically shown in Fig 1.

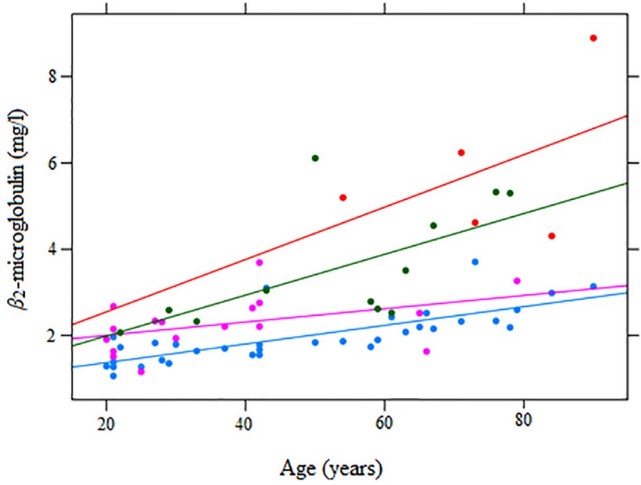

**Fig 1. $\beta_2$-m levels in Covid-19 on admission to the hospital.** Serum $\beta_2$-m levels (mg/l) measured at the time of first SARS-Cov-2 detection are shown in relation to age in a group of 17 patients with mild (magenta ●), 12 patients with moderate (green ●) and five patients with severe Covid-19 (red ●) and 34 approximately age-sex matched uninfected controls (light blue ●). Each dot represents one participant. Linear regression lines of $\beta_2$-m levels on age per group are also included.

**Table 2. Group comparison of reported predictors of outcome including lymphocyte subsets in mild (A) *versus* moderate (B) *versus* severe (C) Covid-19 on admission to the hospital.**

| | Controls | | Mild (A) | | Moderate (B) | | Severe (C) | | |
| | n = 34 | | n = 17 | | n = 12 | | n = 5 | | |
| | Mean | ±SD | Mean | ±SD | Mean | ± SD | Mean | ± SD | Reference range |
|---|---|---|---|---|---|---|---|---|---|
| Age | 48.2 | 21.6 | 36.9 | 18 | 53.3 | 18.1 | 74.4 | 13.8 | |
| WBC x $10^9$/l | 6.6 | 2.1 | 5.1 | 1.7 | 4.9 | 1.7 | 5.9 | 2.0 | 3.9–11.0 |
| $CD3^+$ T cells/µl | 1766 | 520 | 1344 | 629 | 1035 | 255 | 930 | 390 | 1500–4300 |
| $CD3^+$ $CD4^+$ T cells/µl | 1074 | 378 | 780 | 354 | 649 | 144 | 539 | 219 | 322–1750 |
| $CD3^+$ $CD8^+$ T cells/µl | 654 | 296 | 513 | 312 | 367 | 124 | 358 | 261 | 338–1086 |
| $CD4^+$/$CD8^+$ ratio | 1.9 | 0.8 | 1.8 | 1.1 | 1.9 | 0.9 | 1.9 | 0.9 | 0.8–2.4 |
| CD19+ B cells/µl | 273 | 135 | 201 | 140 | 140 | 72 | 342 | 607 | 67–555 |
| NK cells/µl | 238 | 141 | 132 | 67 | 171 | 92 | 197 | 139 | 150–645 |
| Ferritin µg/l | 137 | 162 | 205 | 253 | 254 | 199 | 258 | 184 | f:13–150; m:30–400 |
| D-dimer µg/ml | 0.47 | 0.49 | 0.45 | 0.42 | 0.39 | 0.28 | 1.18 | 1.0 | <0.5 |
| CRP mg/l | 5.31 | 8.35 | 7.83 | 16.6 | 21.6 | 25 | 70.3 | 102.6 | <10 |
| eGFR ml/min/1.73m$^2$ | 104 | 21 | 107 | 16 | 101 | 23 | 64 | 24 | >90 |
| $\beta_2$-m mg/l | 1.98 | 0.61 | 2.27 | 0.64 | 3.57 | 1.39 | 5.85 | 1.86 | 0.8–2.2 |

Means ± SD for age, counts of WBC, $CD3^+$ T lymphocytes, $CD3^+CD4^+$ T cells, $CD3^+CD8^+$ T cells, $CD19^+$ B cells, NK cells, ferritin, D-dimers, CRP, eGFR (CKD-EPI) and $\beta_2$-m are shown in a group of 34 age-sex matched controls and a group of 17 patients with mild (A), 12 patients with moderate (B) and five patients with severe Covid-19 (C). Apart from age and $\beta_2$-m levels, none of the variables was statistically different between the three severity groups by ANOVA test for linear trend (significance level = 0.05).

Our analysis suggests that $\beta_2$-m levels appear to be more proximate causes or correlates of disease severity than age or renal function [47], as stepwise (backward selection) ordinal logistic regression of group on age, sex, eGFR, 10log (CRP) and $\beta_2$-m identified $\beta_2$-m as the only significant predictor. The odds ratio of having severe disease *versus* mild/moderate disease per unit (mg/l) $\beta$2-m was estimated at 4.22 (95% CI:1.86–12.94). Based upon other reports we also measured the following parameters which have been implicated in disease severity: counts of WBC, lymphocytes and $CD8^+$ T cells as well as serum levels of ferritin, D-dimer and CRP. Table 2 summarizes the values of these parameters and other lymphocyte subsets in the different groups. No variable except for age and $\beta_2$-m levels was statistically different between the three severity groups, as calculated by ANOVA (significance level = 0.05).

## Discussion

In a cohort of 34 patients (32 Arabs, two Europeans), who were recently diagnosed and admitted with Covid-19, increased serum levels of $\beta_2$-m were found at the time of presentation to the hospital, a hitherto unreported abnormality [48]. As of February 2021, a PubMed search using the term "$\beta_2$-m AND Covid-19" has resulted in three hits. In two reports, $\beta_2$-m level was measured and found raised in the cerebrospinal fluid of patients with Covid-19-related encephalitis [49, 50], while in the third communication, measurement of $\beta_2$-m was used as human cellular control in Covid-19 testing in samples from the respiratory tract [51]. The results of our study were predictable as other viral infections were also accompanied by increases in $\beta_2$-m concentrations [31]. However, higher levels were noted in those participants who had severe disease with viral pneumonia, resulting in respiratory failure, invasive mechanical ventilation and death in some. This clinical observation induced us to compare patients with mild disease with those who had moderate or severe disease. A statistically significant difference in mean $\beta_2$-m levels was found between the three groups (2.27 ±0.64 mg/l *versus* 3.57

±1.39 mg/l *versus* 5.85±1.86 mg/l) suggesting that severe Covid-19 is associated with preceding higher circulating $\beta_2$-m levels. Age, which consistently represents the strongest risk factor for adverse outcomes in other populations, was also associated with severity (mean age 36.9±18.0 in mild, 53.3±18.1 in moderate and 74.4±13.8 in severe disease), and there was a significant correlation between age and $\beta_2$-m levels. In addition, another observation deserves attention, namely the point in time when raised $\beta_2$-m levels were first determined, which was within the first 48 hours after diagnosis. The time between the first measurement of $\beta_2$-m levels and critical clinical deterioration, *i.e.* mechanical ventilation and transfer to ICU, was shortest (24–48 hours) in the two patients who had the highest $\beta_2$-m levels in this cohort and later died, whereas we observed a time lag of seven, eight and 11 days, respectively, between first $\beta_2$-m measurement and deterioration of respiratory function requiring ICU transfer in the other three patients with severe Covid-19 who survived, favoring initial $\beta_2$-m levels as a predictor not only of disease severity but also outcome. In contrast, other known parameters associated with disease severity and outcome that were measured at admission, such as counts of lymphocytes and CD8[+] T cells, levels of ferritin, D-dimer, CRP and eGFR did not reach statistically significant differences between groups in our analysis, corroborating a potential unique role of $\beta_2$-m in risk assessment very early in the disease course.

In our study, a significant correlation was found between $\beta_2$-m levels and renal function (measured by eGFR). Both parameters are univariately significant correlates of disease severity. In multivariate analysis, however, it was $\beta_2$-m rather than eGFR that explained disease severity. To explain this, we surmise that the kidney could, beside its pivotal role in the metabolism of $\beta_2$-m, become a source of $\beta_2$-m in Covid-19 for various reasons, including the fact that SARS-CoV-2 has demonstrated an exquisite tropism for the kidney since its receptor, membrane-bound angiotensin-converting enzyme 2 (ACE2), is highly expressed in the brush border of proximal tubular cells, and to some extent in podocytes, but neither in endothelial nor mesangial cells of the glomerulus [52, 53]. It would be interesting to study the urinary excretion of $\beta_2$-m in patients with Covid-19, in order to better understand and assess the renal responses to such an intricate viral infection [54–56].

Given the obvious limitations of a single-center study and the small sample size, larger cohorts, preferably in areas of the world other than the Arabian Peninsula, are needed to definitively assess the value of $\beta_2$-m levels as an independent biomarker of disease severity and predictor of outcomes with the advantage of having less fluctuations or extraneous influences than other parameters, such as iron stores for ferritin, coagulopathies for D-dimers and secondary bacterial infections for CRP.

$\beta_2$-m level measurements on presentation, possibly in combination with lymphocyte counting and differentiation as described by others, could be useful to foretell the short-term outcome of Covid-19 [57]. This seems important as the progression to acute respiratory failure commonly occurs rapidly within days after disease onset. Testing this hypothesis in different cohorts seems warranted as the result could facilitate early risk stratification, and thereby optimize the timing of hospitalization for close monitoring and timely therapeutic interventions [58–62].

## Supporting information

**S1 File. DataSetCovid_Saudia.** Clinical, laboratory and radiographic data were handled *via* the electronic medical records system (PowerChart; Cerner, USA), and included participants' demographic details, vital signs, admission notes with chief complaint(s), history of present illness, previous diagnoses, medications, laboratory tests, radiography, and progress notes. These data were extracted from the electronic medical records, deposited and further processed

using REDCap. Within this project, each participant patient was assigned a unique research-specific ID number that was password-protected and accessible to one of the investigators (R.S.A). Data were exported from REDCap into a Microsoft Excel spreadsheet as "Raw Covid Data". With regard to age-sex matched uninfected controls only laboratory data were extracted from electronic medical records and entered separetely into the Excel spreadsheet as"Raw Control Data". "Raw Covid Data" and "Raw Control Data" were used for statistical analysis. (XLSX)

**S1 Table. Covid-19 patients' individual information.** Demographic data, comorbidities, symptoms at presentation and medications are shown for each patient enrolled in the study. (XLSX)

## Author Contributions

**Conceptualization:** Walter Conca.

**Data curation:** Walter Conca, Mayyadah Alabdely, Faisal Albaiz, Michael Warren Foster, Maha Alamri, Nicolaas Nagelkerke, Reem Saad Almaghrabi.

**Formal analysis:** Nicolaas Nagelkerke.

**Investigation:** Walter Conca, Mayyadah Alabdely, Faisal Albaiz, Morad Alkaff, Futwan Al-Mohanna, Nicolaas Nagelkerke.

**Methodology:** Morad Alkaff.

**Project administration:** Reem Saad Almaghrabi.

**Supervision:** Walter Conca, Nicolaas Nagelkerke.

**Visualization:** Nicolaas Nagelkerke.

**Writing – original draft:** Walter Conca.

**Writing – review & editing:** Walter Conca, Michael Warren Foster, Futwan Al-Mohanna, Nicolaas Nagelkerke, Reem Saad Almaghrabi.

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
