## [Decision Letter · Decision Letter 0]

7 Sep 2020

PONE-D-20-14213

Serum β2-microglobulin levels in Coronavirus Disease 2019 (Covid-19): another prognosticator of disease severity?

PLOS ONE

Dear Dr. Conca,

Thank you for submitting your manuscript to PLOS ONE. After careful consideration, we feel that it has merit but does not fully meet PLOS ONE’s publication criteria as it currently stands. Therefore, we invite you to submit a revised version of the manuscript that addresses the points raised during the review process.

Though the topic is interesting, major issues are needed. A  cohort of control subjects should be added due to the small number of patients. References should be updated, cited and discussed. 

We look forward to receiving your revised manuscript.

Kind regards,

Chiara Lazzeri

Academic Editor

PLOS ONE

Journal Requirements:

2. Thank you for your ethics statement : "The study was approved by the Hospital’s Ethics Committee (RAC

Nr: 2201052) and written consent was obtained from all patients"

Reviewers' comments:

Reviewer's Responses to Questions

**Comments to the Author**

1. Is the manuscript technically sound, and do the data support the conclusions?

Reviewer #1: Partly

2. Has the statistical analysis been performed appropriately and rigorously? 

Reviewer #1: Yes

3. Have the authors made all data underlying the findings in their manuscript fully available?

Reviewer #1: Yes

4. Is the manuscript presented in an intelligible fashion and written in standard English?

Reviewer #1: Yes

5. Review Comments to the Author

Reviewer #1: In this manuscript the authors evaluate the potential role of serum b2 microglobulin as a prognostic indicator of COVID-19 severity. Specific points:

• The introduction should be updated, since COVID-19 is a pandemic and not confined to China and few other countries

• In the results section, when describing patients’ characteristics, the authors should also include percentages of symptoms, not only absolute numbers of patients.

• A main table describing demographic and clinical characteristics of each patient would be helpful, rather than a list of comorbidities and medications.

• The authors declare that laboratory tests were performed within 2 days from hospital admission. It is not clear to me which is the time span between laboratory tests and disease progression (i.e. mechanical ventilation, ICU transfer, death). In other words, elevated b2 microglobulin levels are a predictive factor or reflect a different disease status?

• In addition to reference ranges, it would be helpful to also include a cohort of age and sex matched healthy donors to be tested for all the laboratory parameters.

• Given the small size of the cohorts, data in fig.1 would be better represented with individual data points. Moreover, also data reported in Table 1 should also be represented in a figure as individual data points, instead than in a table.

• Although not statistically significant in this cohorts, CRP levels seem higher in severe patients. Is there any correlation with b2 microglobulin levels?

• In the result section the Authors state that the "impairment of renal function did not account for elevated β2-m levels [24]". Eventually, this sentence should be included in the discussion section and not the result section. Nonetheless, both in acute kidney injury and chronic kidney disease serum beta2 microgloubulin levels can be elevated. Among the clinical features of mild and severe COVID-19 patients serum creatinine levels within 2 days from admission should be presented as well as the glomerular filtration rate (calculated with Cockroft-Gault, MDRD or CKD-EPI equation). Correlation between these two parameters and serum beta2microglobulin should be evaluated.

• Is it possible to retrieve data on serum lactate dehydrogenase levels (within 2 days from hospital admission) of the enrolled patients?

6. PLOS authors have the option to publish the peer review history of their article (what does this mean?). If published, this will include your full peer review and any attached files.

Reviewer #1: No

---

## [Author Response · Author response to Decision Letter 0]

9 Feb 2021

Academic Editors

PLoS One

 Riyadh, 9 February 2021

PONE-D-20-14213

Dear Editors,

Please find attached the revision of a manuscript entitled:

“β2-microglobulin levels in Covid-19: another prognosticator of disease severity?” 

by Walter Conca, Mayyadah Alabdely, Faisal Albaiz, Michael W Foster, Maha Alamri, Morad Alkaff, Futwan Al-Mohanna, Nicolaas Nagelkerke, and Reem S Almaghrabi, all at King Faisal Specialist Hospital & Research Centre in Riyadh, Saudi Arabia, except for now retired Prof. emeritus Nicolaas Nagelkerke.

We re-submit a revised version, addding another co-author (Dr Foster) and include answers and comments addressing each point of criticism raised earlier. The answers are as follows:

1. A cohort of control subjects should be added due to the small number of patients. References should be updated, cited and discussed. 

A cohort of age-sex matched uninfected controls was included. More references are cited and discussed.

2. Thank you for your ethics statement: "The study was approved by the Hospital’s Ethics Committee (RAC Nr: 2201052) and written consent was obtained from all patients "Please amend your current ethics statement to include the full name of the ethics committee/institutional review board(s) that approved your specific study.

The full name of the institutional review board that has approved the study is: “Research Advisory Council (RAC)” and was included in the Methods section, as well as in “Ethics statement”.

Reviewer #1

Comments to the Author

Reviewer #1: In this manuscript the authors evaluate the potential role of serum b2 microglobulin as a prognostic indicator of COVID-19 severity. Specific points:

• The introduction should be updated, since COVID-19 is a pandemic and not confined to China and few other countries.

Introduction was thoroughly revised, updated and references added accordingly.

• In the results section, when describing patients’ characteristics, the authors should also include percentages of symptoms, not only absolute numbers of patients.

All patients’ characteristics are included in Table 1, and percentages were added to absolute numbers.

• A main table describing demographic and clinical characteristics of each patient would be helpful, rather than a list of comorbidities and medications.

Table 1 in the main text combines demographics, comorbidities, symptoms/signs, medications and vitals on presentation. We also created another table as a supporting information (S2 Table 1) in which variables are shown for each individual participant rather than by category of severity as defined by WHO criteria.

• The authors declare that laboratory tests were performed within 2 days from hospital admission. It is not clear to me which is the time span between laboratory tests and disease progression (i.e. mechanical ventilation, ICU transfer, death). In other words, elevated b2 microglobulin levels are a predictive factor or reflect a different disease status?

This is an excellent point which we are able to solve only partially. The reason is that in the severe disease group we followed five patients only. However, among these five patients two became critical within 24-48 hours, whereas the other three deteriorated after several days (seven, eight, eleven) supporting the hypothesis that indeed the levels of β2-m have the potential to predict progression as well as outcome. Therefore, a fair answer to the question raised would be that raised β2-m levels on presentation – in our study ≥4 mg/l – are either reflecting critical disease or indicative of progression (“warning sign”) to critical illness at a later point in time. 

• In addition to reference ranges, it would be helpful to also include a cohort of age and sex matched healthy donors to be tested for all the laboratory parameters.

This was done for most parameters on an approximately age-sex matched, uninfected control group and variables were determined for comparative purposes (included in Fig 1 and Table 2).

• Given the small size of the cohorts, data in fig.1 would be better represented with individual data points. Moreover, also data reported in Table 1 should also be represented in a figure as individual data points, instead than in a table.

Fig 1 was re-designed with individual data points according to disease severity categories and linear regression lines were added. We find it hard to represent variables in Table 2 as figure, discussed options and we could not find a reasonable solution to this request.

• Although not statistically significant in this cohorts, CRP levels seem higher in severe patients. Is there any correlation with b2 microglobulin levels?

This is another good point. CRP is also a significant predictor of outcome. It is however correlated with β2-m and in a multivariate analysis that includes both variables, CRP “drops out”. So it appears to contribute little information in contexts when β2-m is measured. The causal pathophysiologic structure behind theses associations needs further study. 

• In the result section the Authors state that the "impairment of renal function did not account for elevated β2-m levels [24]". Eventually, this sentence should be included in the discussion section and not the result section. Nonetheless, both in acute kidney injury and chronic kidney disease serum beta2 microgloubulin levels can be elevated. Among the clinical features of mild and severe COVID-19 patients serum creatinine levels within 2 days from admission should be presented as well as the glomerular filtration rate (calculated with Cockroft-Gault, MDRD or CKD-EPI equation). Correlation between these two parameters and serum beta2microglobulin should be evaluated.

This point of relevance was added in Discussion and potential renal mechanisms contributing to β2-m elevation addressed. Renal function was determined as eGFR (CKD-EPI). eGFR values are included in the raw data (S1 DataSetCovid_Saudia.Excel). Both, eGFR and creatinine, are univariately associated with WHO severity (outcome), although only eGFR significantly so. However, in multivariate analysis that includes both, eGFR and β2-m, eGFR “drops out”. Again, definite conclusions about pathophysiologic mechanisms underlying these associations can only be drawn hypothetically.

• Is it possible to retrieve data on serum lactate dehydrogenase levels (within 2 days from hospital admission) of the enrolled patients?

LDH was not measured in the patients enrolled, unfortunately.

The possible role or significance of β2-m in Covid-19 as an independent predictor of disease severity/outcome has as yet not been addressed or published in accessible databases, and so we hope that with the changes made you will consider this paper suitable for publication.

Sincerely,

Walter Conca MD FACP

---

## [Editor Report · Decision Letter 1]

15 Feb 2021

Serum β2 -microglobulin levels in Coronavirus Disease 2019 (Covid-19): another prognosticator of disease severity?

PONE-D-20-14213R1

Dear Dr. Conca,

We’re pleased to inform you that your manuscript has been judged scientifically suitable for publication and will be formally accepted for publication once it meets all outstanding technical requirements.

Kind regards,

Chiara Lazzeri

Academic Editor

PLOS ONE
---

## [Editor Report · Acceptance letter]

18 Feb 2021

PONE-D-20-14213R1 

Serum *β*_2_-microglobulin levels in Coronavirus Disease 2019 (Covid-19): another prognosticator of disease severity? 

Dear Dr. Conca:

I'm pleased to inform you that your manuscript has been deemed suitable for publication in PLOS ONE. Congratulations! Your manuscript is now with our production department. 

Kind regards, 

on behalf of

Dr. Chiara Lazzeri 

Academic Editor

PLOS ONE